# Improving OOD Robustness via Background-Aware Test-Time-Augmentation in Black-Box and Resource Constrained Settings

**Ping Song**                                                                   *P.Song1@universityofgalway.ie*
*Research Ireland Centre for Research Training in Artificial Intelligence,*
*Data Science Institute, University of Galway, Ireland*

**Adebeoge Ojo**                                                               *Adegboyega.Ojo@carleton.ca*
*School of Public Policy & Administration (SPPA), Carleton University, Canada*

**Edward Curry**                                                         *Edward.Curry@universityofgalway.ie*
*Insight Research Ireland Centre for Data Analytics,*
*Data Science Institute, University of Galway, Ireland*

**Reviewed on OpenReview:** *https://openreview.net/forum?id=xptPQVCy5X*

## Abstract

Deep learning models for text classification typically achieve strong performance on in-distribution (ID) data but often fail to generalize to out-of-distribution (OOD) inputs. This degradation frequently arises because models rely on spurious background cues (e.g., specific syntax or register) learned during training, which become unreliable when the domain changes. While recent Test-Time Augmentation (TTA) approaches have enabled robustness in black-box settings, they often rely on unconstrained rewriting strategies. For instance, standard In-Context Rewriting (ICR) instructs Large Language Models (LLMs) to modify input details to match ID exemplars, creating a high risk of semantic drift and label flipping, particularly when using smaller, resource-constrained LLMs. In this work, we propose a Background-Aware TTA (BA-TTA) framework that strictly disentangles style from semantics. Unlike prior methods that encourage broad paraphrasing, we utilize a semantic-constrained alignment strategy that enables small, efficient LLMs to transform specific background attributes, such as tone and sentence structure, to match in-distribution priors while explicitly enforcing the preservation of original meaning. This approach mitigates OOD degradation by neutralizing spurious background shifts, allowing frozen black-box models to process inputs in their native distribution without risking semantic corruption. Empirical evaluations across multiple text classification benchmarks demonstrate that our targeted alignment strategy outperforms unconstrained augmentation baselines. By generating higher-fidelity augmentations, our method achieves superior OOD robustness with reduced computational overhead, establishing a viable path for deploying robust in resource-limited black-box environments. We validate the versatility of BA-TTA using a range of open-weights generators, from Llama-2 based models to the recent Llama-3.1-8B and Qwen-2.5-7B, showing consistent gains across model families. Our code is available at https://github.com/Ping-Song/OOD_TTA.

## 1 Introduction

Deep learning models for text classification are often expected to perform reliably not only on in-distribution (ID) inputs drawn from the training distribution, but also on unseen out-of-distribution (OOD) data encountered in deployment Hupkes et al. (2023). Achieving such robustness is essential for building safe and

trustworthy NLP systems Song et al. (2026), especially in application domains where errors can be costly, such as spam detection, toxicity monitoring, and healthcare diagnostics. However, OOD generalization in NLP remains a persistent challenge due to the inherently diverse and dynamic nature of language. Real-world data is subject to continuous distributional shifts, where changes in topics, tones, dialects, or writing styles can substantially alter the input distribution. For instance, a sentiment classifier trained on formal movie reviews may fail catastrophically when applied to casual tweets, even if the underlying sentiment is identical. When these background characteristics change at test time, predictions can become unreliable because the model often relies on spurious correlations Sagawa et al. (2020) (e.g., associating specific syntax with a label) rather than robust semantic features.

While a rich body of prior work aims to improve OOD robustness, most existing methods depend on white-box access to model parameters. Techniques such as fine-tuning Bommasani et al. (2021), gradient-based adversarial training, domain-invariant representation learning Nguyen et al. (2021) or uncertainty calibration require modifying the model weights or accessing internal gradients Yuan et al. (2023a). These requirements are often unrealistic in many deployment scenarios. In many practical applications, models are deployed as immutable components on edge devices, or accessed via query-only interfaces (e.g., proprietary large language model APIs) where weights are hidden. Under these "black-box" constraints, parameter-centric robustness techniques are fundamentally inapplicable. Consequently, recent work has explored Test-Time Augmentation (TTA) as a practical alternative O'Brien et al. (2024); Cohen & Giryes (2024). By generating multiple variants of a test input and aggregating their predictions, TTA attempts to bridge the domain gap at inference time. However, conventional TTA methods relying on simple word- or phrase-level perturbations (e.g., synonym replacement) often fail to produce sufficiently rich transformations to correct complex stylistic shifts.

To address these limitations, recent approaches like LLM-TTA have leveraged Large Language Models (LLMs) to perform In-Context Rewriting (ICR). These methods prompt an LLM to rewrite OOD inputs to statistically resemble a set of in-distribution exemplars. While effective at masking distribution shifts, these methods typically rely on unconstrained rewriting strategies. For example, standard ICR prompts often instruct the LLM to "modify details if necessary" to match the training style. This introduces two critical limitations. First, unconstrained rewriting creates a high risk of semantic drift, where the LLM, prioritizing style alignment over content fidelity, accidentally alters the class label (e.g., hallucinating positive sentiment to match a positive ID exemplar). Second, to mitigate the noise introduced by these hallucinations, these methods must rely on the statistical power of large ensembles (aggregating predictions over many augmentations), which strains the computational budget in resource-constrained deployment scenarios.

A primary goal of test-time augmentation is to provide an immediate, "plug-and-play" solution for robust inference. However, for this to be viable in real-world applications, such as on-premise servers or latency-sensitive edge devices, the augmentation process itself must be computationally efficient. While Large Language Models provide the necessary flexibility for domain transformation, the shift toward resource-constrained deployment necessitates strategies that can function effectively with smaller, open-weights models (e.g., the 7B–8B parameter Llama and Qwen families). We demonstrate that by introducing explicit semantic constraints, we can focus the model's limited capacity purely on background alignment. This allows modern, efficient architecture to produce high-fidelity, label-preserving transformations that traditionally would be expected only from much larger architectures.

In this work, we propose a BA-TTA framework (see Figure 1) grounded in the observation that domain shift can often be decomposed into two distinct axes: semantic shift (changes in meaning) and background shift (changes in style/syntax) Arora et al. (2021); Yuan et al. (2023a). We identify that OOD performance degradation is frequently driven by models' reliance on spurious background cues, such as domain-specific phrasing or sentence length, hich become unreliable at test time even when the semantic signal remains unchanged. To mitigate this, we move beyond the "blind" rewriting of prior work and introduce a semantic-constrained alignment strategy. Instead of asking an LLM to broadly "paraphrase" the input, we prompt small, efficient LLMs to align only the specific background characteristics (e.g., tone, register) of OOD inputs to the ID distribution, while preserving label-relevant semantic content.

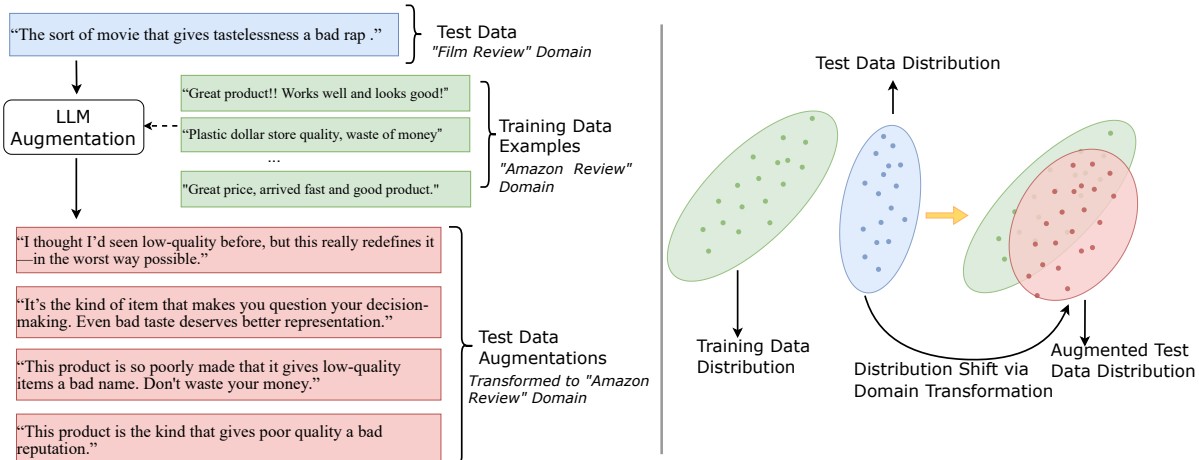

Figure 1: **Test-Time Augmentation with Domain Transformation:** LLM-based test-time augmentation via domain transformation. OOD input and ID examples are fed into LLM with background shift prompts. We show that this process can shift OOD distribution closer to ID distribution.

We validate our approach across seven open-source datasets and one new synthetic benchmark, spanning sentiment analysis, toxicity detection, and topic classification. Our experiments demonstrate that by explicitly correcting background shift, we can achieve superior robustness compared to generic LLM-based rewriting. Crucially, our method demonstrates that strict semantic constraints allow smaller, less capable LLMs to perform effective augmentation, achieving state-of-the-art robustness even when using identical model sizes and inference budgets as baseline methods. Furthermore, we show that our framework remains highly effective across multiple generations of LLMs, including the recent Llama-3.1-8B and Qwen-2.5-7B.

Our contributions are threefold:

1. **Decomposition of Domain Shift**: We empirically demonstrate that significant performance degradation stems from spurious background features (style, syntax) rather than semantic content, identifying "background alignment" as the optimal intervention for OOD robustness.

2. **Semantic-Constrained Rewriting**: We introduce a black-box augmentation framework that uses targeted constraints to rewrite OOD inputs. Unlike standard ICR which risks label flipping, our method strictly preserves task-relevant semantics, enabling the safe use of small LLMs.

3. **Practicality in Resource-Constrained Settings**: We demonstrate the effectiveness of our framework using a diverse range of 7B–8B parameter open-weights models, including Stable Beluga 2, Llama-3.1-8B, and Qwen-2.5-7B. Our results show that targeted background alignment is a highly efficient path to robustness, allowing practitioners to achieve significant OOD gains on consumer-grade hardware without the need for massive, proprietary APIs or high-latency ensembles.

## 2 Related Work

**Test-Time Adaptation.** Test-time adaptation refers to methods that update model parameters or intermediate representations during inference in response to distribution shift (Liang et al., 2025; Yuan et al., 2023b). Unlike conventional domain adaptation, these methods typically do not assume access to labeled target-domain data and are designed to operate under inference-time constraints. Early work in this area focused on adapting batch-normalization statistics or optimizing auxiliary self-supervised objectives at test time. For example, Sun et al. (Sun et al., 2020) proposed test-time training (TTT), where an auxiliary self-supervised task is optimized during inference to refine feature representations. Wang et al. (Wang et al., 2020) introduced TENT, which updates only batch-normalization parameters by minimizing prediction entropy on unlabeled test samples. Subsequent work explored feature alignment (Wang et al., 2023; Jung

|  |  | In-Distribution | Out-of-Distribution (OOD) | | |
| --- | --- | --- | --- | --- | --- |
| Model | Training Source | Amazon (Test) | SST-5 | Dynasent | SemEval |
| T5 | Amazon | 90.11% | 76.12% | 47.73% | 50.07% |
| BERT | Amazon | 90.38% | 68.47% | 42.71% | 44.97% |
| Llama-2 | Amazon | 97.15% | 74.16% | 42.73% | 39.21% |

Table 1: Model performance on Sentiment Analysis, comparing In-Distribution (ID) baselines with Out-of-Distribution (OOD) robustness.

|  |  | In-Distribution | Out-of-Distribution (OOD) | |
| --- | --- | --- | --- | --- |
| Model | Training Source | Civil Comments (Test) | ToxiGen | Adv. Civil |
| T5 | Civil Comments | 90.57% | 65.78% | 46.97% |
| BERT | Civil Comments | 88.46% | 66.74% | 30.46% |
| Llama-2 | Civil Comments | 97.71% | 65.57% | 27.67% |

Table 2: Model performance on Toxicity Detection, comparing In-Distribution (ID) baselines with Out-of-Distribution (OOD) robustness results.

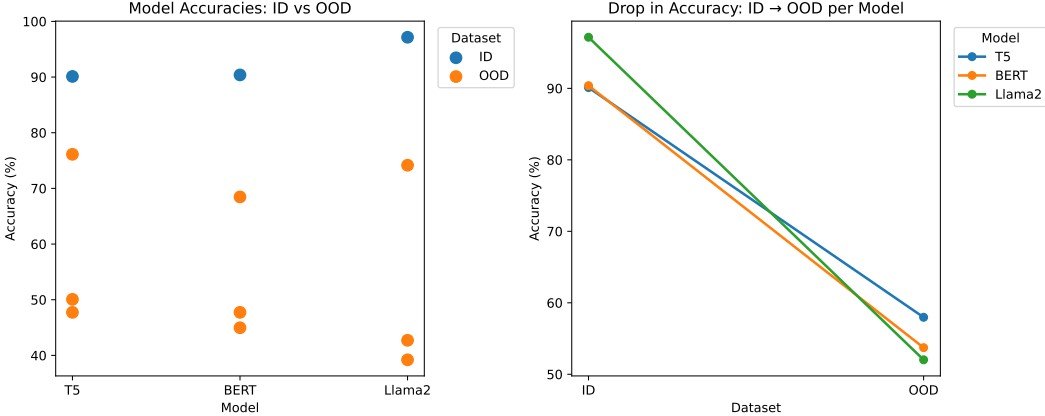

Figure 2: **ID → OOD Performance Gap:** While fine-tuned LLMs achieve high accuracy on ID data, they experience substantial drops when evaluated on OOD datasets, highlighting a critical generalization dilemma.

et al., 2023), domain-specific normalization layers (Wu et al., 2024; Zhang et al., 2023), and meta-learning strategies that prepare models for rapid adaptation under shift (Liang & Chen, 2025). While effective, these methods require white-box access to model internals, careful optimization, and additional stability considerations under severe OOD shift.

**Test-Time Augmentation.** Test-time augmentation extends the idea of data augmentation from training to inference by creating multiple perturbed variants of an incoming sample and aggregating the resulting predictions (Zhang et al., 2022; O'Brien et al., 2024). In vision, these perturbations often include cropping, flipping, color jittering, or geometric transformations (Shorten & Khoshgoftaar, 2019). In text, common strategies include back-translation (Sennrich et al., 2015), synonym replacement, and paraphrasing (Wieting et al., 2017). The goal is to smooth predictions across equivalent views of the input and thereby reduce sensitivity to distributional shifts. Some work further combines augmentation with uncertainty-aware aggregation or calibration (Kumar et al., 2019), although this can become computationally expensive due to repeated forward passes.

Despite its simplicity, standard test-time augmentation may be insufficient under severe OOD shifts if the generated perturbations do not reflect the actual factors of variation in the target domain. This limitation is particularly pronounced in NLP, where domain shifts often involve complex changes in style, discourse structure, topic framing, or pragmatic context rather than local lexical variation alone. Our work belongs to this augmentation-based line, but differs from prior approaches by explicitly targeting *background-feature alignment* while constraining semantic preservation, rather than relying on unconstrained perturbations or generic paraphrastic diversity.

A practical source of confusion in the literature is that the abbreviation "TTA" is sometimes used ambiguously to refer either to *test-time adaptation* or to *test-time augmentation*. These are conceptually different. Test-time adaptation modifies the model or its internal state during inference, whereas test-time augmentation leaves model parameters fixed and instead generates multiple transformed views of the same test input. Since our method never updates classifier parameters, gradients, or internal normalization statistics at inference time, it belongs to the latter category. To avoid misunderstanding, we use the term *BA-TTA* specifically to denote *background-aware test-time augmentation*, not parameter adaptation. In practical deployments, this distinction matters because augmentation-based methods preserve compatibility with frozen black-box models, while adaptation-based methods generally require white-box access and may introduce risks such as unstable online updates, optimizer sensitivity, or interference across sequential test instances. Accordingly, throughout this paper we explicitly separate these two paradigms and position our method as an augmentation-based black-box robustness strategy.

**Augmentation for Text** Text-based models also face OOD challenges, including shifts in vocabulary, style, or topic. Text augmentation at training time, such as synonym replacement, back-translation, or paraphrasing Sennrich et al. (2015); Wieting et al. (2017), is widely used to improve model generalization. However, fewer works explore augmenting text samples directly at test time.

Existing text augmentation strategies often rely on transformations that preserve semantic meaning but vary the lexical or syntactic form Shorten et al. (2021). For instance, synonyms or short paraphrases can be generated using language models O'Brien et al. (2024), then passed through the model to produce an ensemble of predictions. In the presence of domain shift (e.g., changes in terminologies or introduction of new expressions), these augmentations can expose the model to equivalent meanings expressed in different forms, offering additional robustness.

Effectively applying text augmentations at test time poses unique challenges. Minimal changes in word choice might not suffice if the domain shift significantly alters context or linguistic style. Furthermore, real-time text augmentation can introduce latency overhead and risk generating unnatural phrases that degrade model confidence. Recent studies attempt to mitigate these issues by controlling augmentation quality and relevance through learned policies. Overall, test-time text augmentation remains an area for further exploration, especially in conjunction with adaptive methods designed for OOD scenarios.

**LLMs as Augmentation Engines** Recent advances in large language models (LLMs) have opened new opportunities for test-time adaptation. By conditioning on natural language prompts, LLMs can perform high-quality paraphrasing, style transfer, and contextual rewriting without labeled supervision. These capabilities allow LLMs to serve as powerful black-box augmentation engines, capable of reshaping test-time inputs to better resemble training data. However, existing applications of LLM-based rewriting typically treat it as a generic paraphrasing tool, without a principled framework for targeting the underlying structure of distributional shifts.

**Decomposing Distribution Shifts in Text: Background vs. Semantic Features** Emerging evidence suggests that distributional shifts in NLP are governed by two distinct and often orthogonal components Arora et al. (2021):

• Background Shifts: These involve changes in surface-level or contextual features, such as domain (e.g., news vs. social media), style (formal vs. informal), register, or vocabulary. Such shifts often preserve the core semantic content but alter the distributional characteristics that models rely on during training.

| LLM-TTA: In-Context Rewriting (ICR) | BA-TTA: Domain Transformation |
|---|---|
| ### Instructions ###
The assistant is to paraphrase the input text as if it was one of the examples. Change the details of the text if necessary.

### Style Examples ###
<style_transfer_exemplars>

### Input Text ###
Now paraphrase ```"{input_sentence}"``` as if it was one of the examples. Change the details of the text if necessary.

Return the text in the format: ```Paraphrased Text```

### Paraphrased Text ###
Paraphrased Text: | ### Instructions ###
Rewrite the input sentence into the <ID domain>, adjusting only background features such as writing structure, sentence flow, and discourse style.

CRITICALLY:
- Preserve all toxic, abusive, insulting, or hateful expressions EXACTLY.
- Do NOT soften, weaken, sanitize, or remove any toxic or profane words.
- Do NOT replace slurs or insults with milder alternatives.
- Do NOT change the target of the toxicity.
- Do NOT add meta-language about toxicity (e.g., "this comment is offensive").
- Only modify non-semantic stylistic aspects (punctuation, connective words, flow, formatting).

###Example Sentences and their domain:###
{examples}

###Input Sentence:
{input_sentence};

### Transformed Text ###
Rewrite the sentence in <ID domain> while keeping all toxic expressions verbatim and maintaining the same level of toxicity. Only shift background writing style. |

Figure 3: **LLM-TTA Prompt Templates.** Left: In-Context Rewriting (ICR) prompt, which instructs the model to paraphrase the input sentence using provided style exemplars and format the output as "Paraphrased Text". Right: Domain Transformation prompt, which directs the model to convert an input sentence into an equivalent expression in a specified target domain, matching style, tone, vocabulary, and structure, while preserving its original meaning. It shows a toxicity-task instantiation of our general BA-TTA prompt template; only the semantic-preservation clause is task-specific.

• Semantic Shifts: These occur when the task-relevant meaning of inputs changes, such as the introduction of new classes in classification tasks (e.g., previously unseen intents or topics). These shifts are more fundamental, as they affect the decision boundary learned by the model.

## 3 Methodology

### 3.1 Preliminary & Problem Definition

Consider a text classification task where each example consists of an input $x \in X$ and an output label $y \in Y$. The input space $X \subset \mathbb{R}^n$ and output space $Y$ contain $K$ classes. We assume access to a training dataset $D_{\text{train}}$ consisting of pairs $(x, y)$ sampled from the training data distribution $p_{\text{train}}(x, y)$. A model $f$ is trained on this dataset to learn a mapping $f : X \to Y$ that generalizes to unseen data.

At test time, however, the model may encounter an input $x' \in X$ drawn from an unknown distribution $p_{\text{OOD}}(x, y)$, which may differ from the original training distribution. The goal is to ensure that the model remains robust and performs well on such out-of-distribution (OOD) data despite potential distributional shifts.

**Decomposition of Distribution Shifts in Text.** As proposed in prior work Arora et al. (2021), any representation of an input $x$ can be decomposed into two independent and disjoint components: background features: $\phi_b(x) \in \mathbb{R}^m$, semantic features: $\phi_s(x) \in \mathbb{R}^n$.

The overall probability distribution of $x$ can be expressed as $p(x) = p(\phi_s(x))p(\phi_b(x))$. Ideally, background features $\phi_b(x)$ should be independent of the label, while semantic features $\phi_s(x)$ should be label-dependent. Formally, for any class label $y \in Y$: $p(\phi_b(x)|y) = p(\phi_b(x))$, $p(\phi_b(x)|y) \neq p(\phi_s(x))$. This distinction allows us to categorize textual distribution shifts into two major types:

- **Background Shift:** Occurs when the domain, style, or context of text changes, even if the underlying semantics remain the same. For example, transitioning from Amazon product reviews to film reviews.
- **Semantic Shift:** Occurs when the meaning or class distribution of text changes, such as the emergence of new or unseen classes during inference.

Our proposed approach aims to mitigate the impact of background shifts while preserving semantic integrity, thereby enhancing OOD robustness.

### 3.2 BA-TTA: Robustness through Test-Time Domain Transformation

**Transformation & Augmentation** To reduce the distribution gap between OOD and ID data, we define an augmentation function $a_z$ that transforms an input $x$ into an augmented version $x'$. Our objective is to decouple the distribution shift into two components:

- Background Alignment: Shift background features to match the ID distribution: $p(\phi_b(x')) \approx p_{ID}(\phi_b(x))$.

- Semantic Preservation: Strictly maintain the semantic features of the original input: $p(\phi_s(x')) \approx p(\phi_s(x))$.

To implement this separation, we employ a semantic-constrained prompting strategy (see Figure 3, Right, right). Unlike standard In-Context Rewriting (ICR), which approximates the target distribution by broadly mimicking exemplars and instructing the model to "change details if necessary," our prompts explicitly instruct the auxiliary LLM to transform specific background attributes (e.g., tone, sentence structure, lexical style, or contextual framing) while preserving equivalent label-relevant semantics.

This explicit decomposition serves two functions. First, it reduces the risk of semantic drift by constraining the rewrite to preserve class-relevant meaning rather than encouraging open-ended paraphrasing. Second, by narrowing the generation objective from unrestricted rewriting to targeted background adjustment, BA-TTA enables smaller, resource-efficient LLMs to produce useful augmentations without requiring the massive parameter counts often associated with safe unconstrained generation.

Importantly, BA-TTA does not assume that the auxiliary 7B–8B model can perfectly translate an OOD input into the full in-distribution manifold. Rather, it assumes that the model can perform *partial but useful* adjustment of background features that reduces domain mismatch without altering the underlying semantic content relevant to the downstream label. In this sense, the method relies on *background correction* rather than exact domain recovery. This assumption is most plausible under moderate domain shifts where style, register, or discourse context are major sources of mismatch. When the shift is more extreme, or when semantic meaning is tightly entangled with surface form, incomplete alignment or semantic drift may still occur, limiting the benefit of BA-TTA.

In practice, however, this alignment may be incomplete. When the domain shift involves short, implicit, or semantically dense inputs, or when stylistic background is tightly entangled with label-relevant meaning, a compact 7B–8B auxiliary model may fail to move the sample sufficiently toward the ID background without introducing semantic distortion. Thus, BA-TTA is best viewed as relying on *approximate* background correction rather than guaranteed domain recovery, and its benefits are expected to diminish under complex or extreme shifts that require stronger semantic control, deeper pragmatic reasoning, or broader domain knowledge.

**Prompt generalization and construction** Although Figure 3 illustrates a toxicity-specific prompt instance, our BA-TTA prompting scheme is not manually engineered from scratch for each dataset. Instead, it follows a reusable two-part template consisting of: (1) a task-agnostic backbone that instructs the auxiliary LLM to rewrite only background attributes (e.g., tone, register, wording, sentence structure, or stylistic context) so that the input better matches the in-distribution prior, while preserving the original meaning and avoiding hallucinated content; and (2) a lightweight task-specific semantic-preservation clause that specifies which label-relevant factors must remain invariant. This clause can be derived directly from the task definition or annotation guideline rather than handcrafted per example. For instance, for sentiment classification,

the constraint is to preserve sentiment polarity, intensity, and target; for toxicity detection, to preserve abusive/toxic intent, severity, and target; and for topic classification, to preserve topic-defining content and category-relevant facts. Thus, applying BA-TTA to a new task does not require designing a new prompt from scratch, but only instantiating a small semantic-preservation slot within a generic prompt template.

**Inference and Aggregation** At inference time, multiple augmented versions of an OOD input $x'$ are generated using the augmentation functions. These augmented inputs are then passed through the model $f$, and their predictions are aggregated to obtain a more robust final prediction.

Let $A(x)$ represent the set of augmented samples generated from $x$. The final prediction $\hat{y}$ is computed as $\hat{y} = \arg\max_y \sum_{x' \in A(x)} p(y|f(x'))$.

This aggregation strategy helps smooth out distributional variances, ensuring that the model predictions remain stable and resilient against OOD variations.

## 4 Experiment Setup

### 4.1 Datasets

To evaluate the effectiveness of our proposed test-time augmentation (TTA) framework, we utilize datasets across three text classification tasks: sentiment analysis, toxicity detection, and news topic classification. These tasks reflect real-world scenarios where models are required to generalize beyond their training distribution. Our selection of in-distribution (ID) and out-of-distribution (OOD) datasets is designed to capture both background shifts (changes in contextual framing) and semantic shifts (variations in meaning), which are the two primary factors contributing to distribution gaps. Please see Table 14 and Table 15 samples from OOD datasets.

**Sentiment Classification** For sentiment classification, we use a benchmark that includes a three-way classification task with labels: positive, neutral, and negative. The ID dataset consists of Amazon reviews McAuley & Leskovec (2013), representing a consumer-driven review domain. The OOD datasets are DynaSent Potts et al. (2020), SST-5 Socher et al. (2013), and SemEval Nakov et al. (2019); Yuan et al. (2023a), introducing variations in linguistic style, domain, and sentiment annotation schemes. These shifts capture both background changes (e.g., different sources of user-generated content) and potential semantic variations (e.g., differences in sentiment annotation granularity).

**Toxicity Detection** The toxicity detection task is framed as a binary classification problem, distinguishing between toxic (positive) and non-toxic (negative) language. The ID dataset is Civil Comments Borkan et al. (2019), a collection of user comments from a moderated online platform. The OOD datasets include an adversarially augmented version of Civil Comments - AdvCivil Borkan et al. (2019), as well as two additional datasets. Here, the OOD evaluation specifically targets adversarial shifts and diverse online environments, challenging the model's ability to decouple toxic intent from benign community slang.

**News Topic Classification** For the topic classification task, we focus on a four-way categorization problem distinguishing between World, Sports, Business, and Science/Technology topics. The ID dataset is AG News, representing a standard distribution of formal journalistic reporting. The OOD dataset is Twitter-Topic. These benchmarks introduce significant background shifts, moving from structured news articles to informal social media posts, effectively testing the model's resilience to changes in register and length while the underlying topical semantics remain consistent.

### 4.2 Baseline Methods

**Word-Level Augmentations** As a representative of heuristic-based TTA, we include word-level perturbation methods widely used in NLP robustness research. Following the implementation by Lu et al. (2022), we utilize the nlpaug library to perform stochastic word insertion and substitution. Specifically, each word in the input text has a 30% probability of being augmented, up to a maximum of 10 words per input. Substitutions are generated using BERT-based contextual prediction to maintain local fluency. However, unlike our method, these perturbations are local and structure-agnostic; they introduce noise to smooth predictions but

fail to systematically address the broader stylistic or domain shifts (e.g., syntax, register) that characterize true distribution mismatch.

**Back-Translation** We also compare against Back-Translation, a standard "whole-text" augmentation strategy often used for paraphrasing. We employ an English ↔ German translation loop, where the input is translated to German and then back to English to generate a paraphrase. While this approach captures some global semantic structure, it functions as a blind paraphraser: it alters the text without any explicit guidance on which background features (e.g., tone, formality) should be aligned to the ID distribution. Consequently, it may either fail to correct the domain shift or accidentally introduce semantic errors during the translation process.

**Standard In-Context Rewriting (ICR)** We select the LLM-based TTA method proposed by O'Brien et al. (2024) as our primary baseline, as it represents the current state-of-the-art for black-box robustness. This approach utilizes In-Context Learning to adapt OOD inputs: the LLM is provided with 16 randomly selected in-distribution (ID) exemplars and instructed to rewrite the test input to resemble them.

Crucially, this baseline employs an unconstrained prompting strategy. As noted in their implementation, the model is instructed to "change the details of the text if necessary" to achieve stylistic alignment. While this generates diverse augmentations that capture the broad "vibe" of the ID data, it prioritizes stylistic mimicry over semantic fidelity. We follow their experimental setup by generating $N$ stochastic rewrites per input and aggregating predictions, providing a direct comparison between their diversity-driven rewriting and our semantic-constrained alignment.

### 4.3 Implementation Details

**Task Models** Task Models We investigate black-box robustness across diverse model architectures to ensure our findings generalize beyond a single paradigm. We select representative models for the three primary architectural families: Encoder-Only: BERT Devlin et al. (2019), using the fine-tuned checkpoints from O'Brien et al. (2024). Encoder-Decoder: T5-Large Raffel et al. (2020) , a generative model optimized for sequence-to-sequence tasks. Decoder-Only: Llama-2 (Touvron et al., 2023), representing the family of causal large language models. This diverse selection allows us to verify that our Background-Aware TTA is agnostic to the underlying classifier architecture. Table 1, Table 2 and Figure 2 shows the ID-OOD performance gap.

**LLM-based Domain Transformation.** To evaluate whether BA-TTA generalizes across modern compact open-weight LLMs, we employ three instruction-tuned 7B–8B augmenters: Stable Beluga 2-7B (SB2) (Mahan et al., 2023), Llama-3.1-8B-Instruct, and Qwen-2.5-7B-Instruct. All three are used as auxiliary domain transformation models under the same BA-TTA prompting framework. Unlike prior work that relies on massive proprietary APIs (e.g., GPT-4-class models), our goal is to study whether semantically constrained test-time transformation can be carried out effectively using locally deployable open models in the 7B–8B range. This multi-augmenter setup allows us to verify that BA-TTA is not tied to a single auxiliary LLM family, but is instead compatible with recent instruction-following open-weight models.

For each test input, we generate $N = 4$ stochastic augmentations using the selected auxiliary LLM and combine them with the original input during aggregation. We provide randomly selected 16 ID exemplars in the context window to guide background alignment toward the training distribution. Using compact 7B–8B instruction-tuned models substantially reduces deployment cost relative to 70B+ or API-based alternatives, while still enabling high-fidelity controlled rewriting under the BA-TTA prompt.

**Resource-constrained setup.** In all main experiments, the auxiliary augmentation model is deployed locally using a standard 16-bit inference configuration (BF16 for Llama-3.1-8B-Instruct and FP16 for Qwen-2.5-7B-Instruct). For each test sample, we generate $N = 4$ stochastic rewrites with 16 randomly sampled ID exemplars in the prompt. Thus, the reported results correspond to a compact local 7B–8B augmentation setup, rather than a large proprietary API or a retraining-based test-time adaptation pipeline. In this work, *resource-constrained* refers to moderately constrained local settings where large proprietary APIs or retraining-based adaptation are impractical, but a compact local 7B–8B instruction-tuned augmenter remains available. Under this setup, the main computational overhead of BA-TTA comes from test-time rewriting,

| Augmentation | Sentiment | | | Toxicity | | | News → Tweets | | |
|---|---|---|---|---|---|---|---|---|---|
| | T5 | BERT | Llama | T5 | BERT | Llama | T5 | BERT | Llama |
| *Standard Baselines* | | | | | | | | | |
| None | 57.97% | 52.05% | 52.04% | 58.89% | 53.91% | 53.65% | 89.01% | 88.57% | 86.42% |
| Word insertLu et al. (2022) | 56.69% | 51.38% | 48.28% | 57.62% | 53.36% | 51.85% | 89.96% | 88.87% | 91.36% |
| Word substituteLu et al. (2022) | 54.57% | 49.93% | 44.25% | 57.00% | 52.96% | 50.37% | 89.18% | 88.51% | 90.43% |
| Back translationSennrich et al. (2016) | 54.16% | 50.05% | 48.45% | 57.70% | 57.13% | 55.80% | 88.41% | 88.41% | 90.37% |
| *Augmentation: Llama-2-7B* | | | | | | | | | |
| In-context rewriting O'Brien et al. (2024) | 59.66% | **56.20%** | 50.50% | 63.76% | 60.84% | 54.88% | 90.38% | 89.51% | 90.05% |
| Domain transformation (Ours) | **59.95%** | 55.94% | **52.20%** | **68.59%** | **66.08%** | **63.54%** | **91.11%** | **90.33%** | **91.53%** |
| *Augmentation: Llama-3.1-8B* | | | | | | | | | |
| In-context rewriting O'Brien et al. (2024) | 59.29% | **55.42%** | 49.65% | 58.26% | 56.59% | 49.98% | 90.08% | 88.88% | 87.29% |
| Domain transformation (Ours) | **60.85%** | 55.10% | **53.43%** | **69.10%** | **67.27%** | **63.19%** | **90.54%** | **90.64%** | **91.64%** |
| *Augmentation: Qwen-2.5-7B* | | | | | | | | | |
| In-context rewriting O'Brien et al. (2024) | 57.17% | 53.98% | 53.25% | 60.30% | 57.84% | 51.71% | 90.84% | **90.88%** | **91.67%** |
| Domain transformation (Ours) | **61.25%** | **58.60%** | **55.20%** | **67.10%** | **66.54%** | **64.75%** | **90.88%** | 90.49% | 91.46% |

Table 3: **Main OOD Performance Results.** We compare standard augmentation baselines against LLM-based strategies across three task model architectures and three different augmentation backbones. Values represent % accuracy.

| Augmentation | Sentiment | | | Toxicity | | | News |
|---|---|---|---|---|---|---|---|
| | SST-5 | Dynasent | SemEval | ToxiGen | Adv. Civil | ImplicitHate | Tweets |
| *Standard Baselines* | | | | | | | |
| None | 72.92% | 44.40% | 44.75% | 66.00% | 35.07% | 65.38% | 88.00% |
| Word insertLu et al. (2022) | 69.93% | 42.65% | 43.77% | 66.70% | 30.78% | 65.35% | 90.06% |
| Word substituteLu et al. (2022) | 64.40% | 41.65% | 42.70% | 64.19% | 30.87% | 65.27% | 89.37% |
| Back translationSennrich et al. (2016) | 67.10% | 41.91% | 43.65% | 64.76% | 41.34% | 64.53% | 89.06% |
| *Augmentation: Llama-2-7B* | | | | | | | |
| In-context rewriting | 72.82% | **47.71%** | 45.83% | 63.35% | 49.88% | **66.26%** | 89.98% |
| Domain transformation (Ours) | **74.75%** | 46.67% | **46.67%** | **66.63%** | **65.94%** | 65.63% | **90.99%** |
| *Augmentation: Llama-3.1-8B* | | | | | | | |
| In-context rewriting | 70.97% | 47.66% | **45.73%** | 61.33% | 39.16% | 64.35% | 88.84% |
| Domain transformation (Ours) | **74.04%** | **48.08%** | 47.26% | **64.86%** | **69.50%** | **65.20%** | **90.97%** |
| *Augmentation: Qwen-2.5-7B* | | | | | | | |
| In-context rewriting | 69.59% | 48.59% | 46.23% | 60.03% | 47.33% | 62.49% | **90.96%** |
| Domain transformation (Ours) | **74.78%** | **51.48%** | **48.79%** | **62.97%** | **71.04%** | **63.38%** | 90.94% |

Table 4: **Main OOD Performance Results.** Detailed comparison across three augmentation backbones. Bold values indicate the best performing method within each backbone group for each dataset.

which scales with the number of augmentations, prompt length, and model precision. We do not interpret this setting as direct evidence of feasibility in ultra-low-resource edge environments.

**Inference and Aggregation Strategy** To mitigate the variance inherent in stochastic generation, we adopt an ensemble aggregation strategy ($N_{total} = 5$, including the original input):

For Discriminative Models (BERT): We average the calibrated softmax probability distributions across all five inputs. The final prediction is determined by the class with the highest average probability: $\hat{y} = \arg\max_c \frac{1}{N} \sum_{i=1}^{N} P(y_c|x_i)$. This leverages the model's confidence scores to smooth out noise.

For Generative Models (T5 & Llama-2): Since these models output discrete text labels, we employ Majority Voting. The final class label is determined by the most frequent prediction among the augmented set. This consensus mechanism ensures that the final decision is robust to outliers or hallucinations in any single augmentation.

# 5 Results

## 5.1 BA-TTA Improves OOD Robustness

The out-of-distribution (OOD) robustness performance on three sentiment benchmarks (SST-5, Dynasent, SemEval), three toxicity detection benchmarks (ToxiGen, Adv. Civil, ImplicitHate), and one news-topic benchmark (Tweets) is evaluated across fine-tuned T5, BERT, and Llama-2 task models. We compare three classes of methods: no augmentation, standard augmentation baselines (word insert, word substitute, and back-translation), and two LLM-based test-time augmentation (TTA) approaches, namely in-context rewriting (ICR) and our proposed domain transformation framework (BA-TTA). In addition to the original SB7 augmentation backbone, we further evaluate BA-TTA with two recent open-weight instruction-tuned generators, Llama-3.1-8B-Instruct and Qwen-2.5-7B-Instruct, in order to test whether the method generalizes across modern 7B–8B augmentation models.

Table 3 summarizes the overall comparison across augmentation backbones, and Table 4 provides a dataset-level breakdown. Overall, BA-TTA achieves the strongest average performance across all three augmentation generators. With SB7, BA-TTA attains average scores of 59.95%, 68.59%, and 91.11% for T5 on sentiment, toxicity, and news, respectively, while the corresponding averages for BERT and Llama-2 remain competitive at 66.08% / 90.33% and 63.54% / 91.53% on toxicity and news. This trend extends to newer augmentation backbones: with Llama-3.1-8B-Instruct, BA-TTA reaches 60.85%, 69.10%, and 90.54% for T5 across sentiment, toxicity, and news, while Qwen-2.5-7B-Instruct yields 61.25%, 67.1%, and 90.88%. Similar gains hold for BERT and Llama-2, showing that the benefits of BA-TTA are not tied to a single auxiliary LLM family, but generalize across recent compact open-weight instruction-tuned models (Tables 3 and 4).

At the individual dataset level, BA-TTA remains particularly effective on sentiment and news benchmarks. For example, under the original SB7 setting (Table 9), BA-TTA improves BERT on SST-5 from 68.47% without augmentation to 73.69%, outperforming ICR at 72.01%. Likewise, for T5 on Tweets, BA-TTA reaches 91.11%, compared with 89.01% without augmentation and 90.38% under ICR. Similar patterns persist with the newer augmentation generators: under Qwen-2.5-7B-Instruct (Table 10), BA-TTA yields 72.11% on BERT for SST-5 and 90.64% on BERT for Tweets, and under Llama-3.1-8B-Instruct (Table 11), BA-TTA reaches 76.40% on T5 for SST-5 and 90.88% on T5 for Tweets. These results indicate that semantically constrained background transformation consistently improves OOD robustness when the distribution shift is associated with style, register, or contextual background differences.

The effect of BA-TTA is especially notable on Adv. Civil, which contains long, context-rich comments drawn from real-world discussions. Under SB7 (Table 9), BA-TTA improves over ICR by +18.69 points for T5 (78.76% vs. 60.07%), +24.15 points for BERT (76.94% vs. 52.79%), and +19.06 points for Llama-2 (58.37% vs. 39.31%). Strong gains on this dataset also persist with newer augmentation generators: for example, BA-TTA reaches 77.31% for T5 with Llama-3.1-8B-Instruct and 72.21% for BERT with Qwen-2.5-7B-Instruct (Tables 11 and 10). This suggests that BA-TTA is particularly effective when OOD examples contain rich background attributes that can be adjusted toward the ID distribution while preserving task-relevant semantics.

At the same time, the benefits are not uniform across all toxicity benchmarks, which helps clarify the distinction between the BA-TTA framework itself and the capability of a particular auxiliary LLM to realize it. Our method does not require the auxiliary 7B–8B model to fully translate an OOD input into the ID distribution, but it does assume that the model can perform sufficient background adjustment to reduce the mismatch while preserving label-relevant semantics (Figure 4). This assumption is clearly satisfied in stronger settings such as SB7 and GPT-4o, but the results on ToxiGen and, in some settings, ImplicitHate show that some compact open-weight augmenters do not reliably realize the intended transformation under more difficult shifts. For example, with Llama-3.1-8B-Instruct, BA-TTA does not improve over the no-augmentation baseline on ToxiGen for any of the three task models, and similar patterns appear for ImplicitHate and for Qwen-2.5-7B-Instruct on selected task-model combinations (Tables 10 and 11). We interpret these cases as evidence of two related backbone-level limitations: *incomplete style alignment*, where the rewrite fails to move the sample sufficiently toward the ID background, and *insufficient transformation ability*, where a compact 7B–8B model cannot safely rewrite inputs whose label semantics are tightly entangled with surface form.

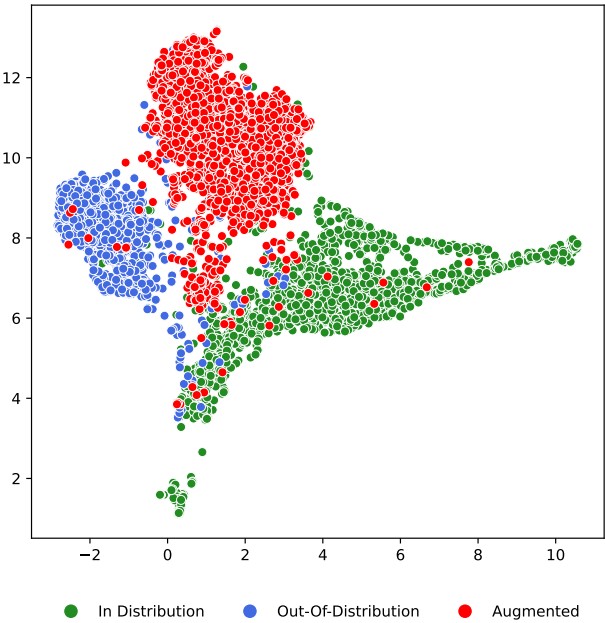

Figure 4: **2D UMAP Projection of Text Embeddings:** visualization of distribution shift and test-time augmentation. Green points represent in-distribution samples, blue points are out-of-distribution samples, and red points are the augmented examples generated at test time. Augmented samples bridge the gap between in- and out-of-distribution regions, illustrating how TTA expands the model's support to improve robustness.

Compared with longer or more stylistically variable inputs, ToxiGen and ImplicitHate often contain short, semantically concentrated, and pragmatically implicit toxic expressions, leaving less editable background context for transformation and making semantic preservation more delicate.

**Failure Analysis.** A qualitative inspection of unsuccessful cases supports this interpretation. We observe three recurring failure modes: (1) *incomplete style alignment*, where the transformed text preserves the original meaning but remains too close to the OOD phrasing to substantially reduce distribution mismatch; (2) *semantic drift*, where the rewrite unintentionally changes sentiment polarity, toxicity intensity, or topic emphasis; and (3) *hallucinated or weakened detail*, where unsupported content is introduced or the original target cue is diluted. These failures occur most often on short, semantically concentrated, and pragmatically implicit inputs such as ToxiGen and ImplicitHate, where there is limited editable background context and even small wording changes can affect the downstream label. Overall, these findings refine our main claim: BA-TTA is most effective when the domain shift is dominated by background/style mismatch and the auxiliary LLM is sufficiently capable of carrying out semantics-preserving transformation. Under more extreme shifts, the limiting factor is not the BA-TTA formulation itself, but whether the selected compact augmentation model has enough semantic control, pragmatic reasoning ability, and domain knowledge to realize the intended transformation faithfully. Representative examples are provided in Appendix D.

## 5.2 Comparison with a Strong Proprietary API Upper Bound

To better validate the resource-constrained motivation, we compare our best local 7B/8B BA-TTA models against a strong proprietary API model, GPT-4o, under both unconstrained in-context rewriting (ICR) and our proposed BA-TTA framework. As shown in Table 9, unconstrained rewriting remains unreliable even for a state-of-the-art API model: GPT-4o (ICR) often underperforms the original baseline, e.g., on T5 it drops from 76.12% to 73.69% on SST5 and from 65.78% to 58.47% on ToxiGen . In contrast, GPT-4o with

| Task Model | Augmentation Strategy | Sentiment | | | Toxicity | | | News |
|---|---|---|---|---|---|---|---|---|
| | | SST5 | DY | SE | TG | AC | IH | Tweets |
| **T5** | Best Local 7B/8B (BA-TTA) | 77.99% | 55.14% | 52.20% | 66.63% | 78.76% | 64.37% | 91.11% |
| | GPT-4o (BA-TTA) | 76.77% | 61.67% | 58.15% | 67.58% | 84.22% | 62.07% | 91.30% |
| | *Performance Gap* | -1.22% | 6.53% | 5.95% | 0.95% | 5.46% | -2.30% | 0.19% |
| **BERT** | Best Local 7B/8B (BA-TTA) | 73.88% | 51.30% | 50.63% | 66.42% | 75.00% | 64.59% | 90.64% |
| | GPT-4o (BA-TTA) | 74.53% | 58.29% | 54.78% | 67.16% | 76.94% | 62.67% | 91.50% |
| | *Performance Gap* | 0.65% | 6.99% | 4.15% | 0.74% | 1.94% | -1.92% | 0.86% |
| **Llama2** | Best Local 7B/8B (BA-TTA) | 74.07% | 48.01% | 43.53% | 66.84% | 57.52% | 67.94% | 91.72% |
| | GPT-4o (BA-TTA) | 75.73% | 55.41% | 53.98% | 65.47% | 58.37% | 67.33% | 91.70% |
| | *Performance Gap* | 1.66% | 7.40% | 10.45% | -1.37% | 0.85% | -0.61% | -0.02% |

Table 5: **Detailed performance gap analysis** of BA-TTA comparing the best local 7B-8B models (including Llama-2-7B, Qwen-2.5-7B, Llama-3.1-8B) against the GPT-4o upper bound, with both utilizing the proposed BA-TTA framework. Positive values indicate a GPT-4o advantage, while negative values (blue) indicate instances where the local 7B/8B model strictly outperforms the massive proprietary API.

| Task Model | Augmentation Strategy | Sentiment | | | Toxicity | | | News | Avg. |
|---|---|---|---|---|---|---|---|---|---|
| | | SST5 | DY | SE | TG | AC | IH | Tweets | |
| **T5** | Original | 76.12% | 47.73% | 50.07% | 65.78% | 46.97% | 62.07% | 88.40% | 62.45% |
| | GPT-4o (ICR) | 73.69% | 54.86% | 50.32% | 58.47% | 53.52% | 61.00% | 91.30% | 63.31% |
| | GPT-4o (BA-TTA) | **76.77%** | **61.67%** | **58.15%** | **67.58%** | **84.22%** | **62.07%** | **91.30%** | **71.68%** |
| **BERT** | Original | 68.47% | 42.71% | 44.97% | 66.74% | 30.46% | **64.33%** | 87.80% | 57.93% |
| | GPT-4o (ICR) | 72.48% | 43.10% | 38.87% | 58.37% | 42.11% | 60.53% | 91.10% | 58.08% |
| | GPT-4o (BA-TTA) | **74.53%** | **58.29%** | **54.78%** | **67.16%** | **76.94%** | 62.67% | **91.50%** | **69.41%** |
| **Llama2** | Original | 74.16% | 42.75% | 39.20% | **65.89%** | 29.73% | **68.80%** | 87.00% | 58.22% |
| | GPT-4o (ICR) | 68.10% | 43.77% | 38.65% | 58.05% | 28.88% | 63.33% | **92.20%** | 56.14% |
| | GPT-4o (BA-TTA) | **75.73%** | **55.41%** | **53.98%** | 65.47% | **58.37%** | 67.33% | 91.70% | **66.86%** |

Table 6: **Performance evaluation of GPT-4o across varying augmentation strategies.** Unconstrained rewriting (ICR) frequently underperforms or marginally improves upon the original baseline, while our semantic-constrained framework (BA-TTA) consistently unlocks the model's potential, indicating that structural constraints are strictly necessary to mitigate semantic drift regardless of model scale.

BA-TTA consistently improves performance, raising the average score from 63.31% to 71.68% for T5, from 58.08% to 69.41% for BERT, and from 56.14% to 66.86% for Llama2. These results show that the gains come primarily from the semantic-constrained BA-TTA formulation rather than from model scale alone, and they confirm that even very strong API models benefit substantially from controlled background-aware transformation.

Table 6 further shows that the performance gap between GPT-4o BA-TTA and the best local 7B/8B BA-TTA model is often modest. For T5, the local model remains highly competitive, trailing GPT-4o by only 1.22% on SST5, 0.95% on ToxiGen , and 0.19% on Tweets, while outperforming it by 2.30% on ImplicitHate. For BERT, the local model surpasses GPT-4o on SST5 and IH by 0.65% and 1.92%, respectively, and stays within 0.74%-6.99% on the remaining datasets. For Llama2, the local model exceeds GPT-4o on TG by 1.37% and on ImplicitHate by 0.61%, while being nearly identical on Tweets (0.02% gap). Although GPT-4o retains a larger advantage on some sentiment benchmarks such as DY and SE, the overall results indicate that a carefully designed local BA-TTA pipeline can recover most of the robustness benefits of a much larger proprietary API. This comparison therefore provides a meaningful upper bound and supports our claim that BA-TTA offers a favorable practicality-performance trade-off in resource-constrained settings.

| Task Model | No aug. | ICR (16b) | BA-TTA (16b) | BA-TTA (8b) | $\Delta$ (8b$-$16b) |
|---|---|---|---|---|---|
| T5 | 76.12% | 70.43% | 76.40% | 77.80% | +1.40% |
| BERT | 68.47% | 67.07% | 73.88% | 73.41% | -0.47% |
| Llama-2 | 74.16% | 71.27% | 74.07% | 73.41% | -0.66% |

Table 7: SST5 OOD performance using `Llama-3.1-8B-Instruct` as the auxiliary augmentation backbone. The table compares no augmentation, 16-bit ICR, 16-bit BA-TTA, and 8-bit BA-TTA. Quantization largely preserves the robustness benefit of BA-TTA on this representative dataset.

## 5.3 Adaptability to Ultra-Low-Resource Environments

To probe whether BA-TTA can move toward lower-resource deployment, we evaluate an 8-bit quantized version of the auxiliary augmentation model on SST5 as a representative case study. As shown in Table 7, 8-bit quantization of the `Llama-3.1-8B-Instruct` augmentation backbone largely preserves the robustness benefit of BA-TTA on SST5, with only small changes relative to the standard 16-bit configuration. The 8-bit variant changes performance by only +1.40 points for T5, -0.47 points for BERT, and -0.66 points for Llama-2. Moreover, under 8-bit inference, BA-TTA continues to outperform unconstrained ICR for all three task models on SST5. These results suggest that quantization is a promising direction for pushing BA-TTA toward lower-resource settings without fully sacrificing transformation quality. At the same time, we interpret this as a proof of concept rather than definitive evidence of suitability for strict sub-16 GB edge deployment, which would require broader evaluation and potentially additional optimizations such as 4-bit inference, prompt compression, or fewer stochastic rewrites.

## 5.4 Practicality and Latency

To assess the practical overhead of BA-TTA, we measure end-to-end *augmentation-side* wall-clock latency under a fixed hardware and decoding configuration, including prompt construction, auxiliary generation, and output cleaning. We compare unconstrained in-context rewriting (ICR) and BA-TTA under matched settings, using the same local augmentation backbone, the same number of stochastic rewrites ($N = 4$), and the same prompt-exemplar budget. Table 8 reports results on two representative OOD settings: SST5 with Llama-3.1-8B-Instruct and Adv. Civil with Qwen-2.5-7B-Instruct.

Overall, BA-TTA introduces only a modest latency overhead relative to ICR. On SST5 with Llama-3.1-8B-Instruct, BA-TTA requires 2.60 seconds per sample on average, compared with 2.18 seconds for ICR, corresponding to a 1.19× overhead. On Adv. Civil with Qwen-2.5-7B-Instruct, BA-TTA requires 5.63 seconds per sample versus 5.01 seconds for ICR, corresponding to a 1.12× overhead. In both cases, the increased latency is partly explained by slightly longer generations under the constrained BA-TTA prompt. These results suggest that BA-TTA improves robustness at a comparable augmentation-time cost, rather than requiring substantially more expensive generation than unconstrained rewriting. At the same time, the absolute latency remains non-trivial, indicating that the current 7B–8B instantiation is better suited to moderately resource-constrained local deployments than to ultra-low-latency edge scenarios.

| Dataset | Augmenter | Method | #Augs | Avg latency/sample (s) | Avg latency/aug (s) | Relative to ICR |
|---|---|---|---|---|---|---|
| SST5 | Llama-3.1-8B-Instruct | ICR | 4 | 2.1806 | 0.5452 | 1.00× |
| SST5 | Llama-3.1-8B-Instruct | BA-TTA | 4 | 2.5984 | 0.6496 | 1.19× |
| Adv. Civil | Qwen-2.5-7B-Instruct | ICR | 4 | 5.0086 | 1.2522 | 1.00× |
| Adv. Civil | Qwen-2.5-7B-Instruct | BA-TTA | 4 | 5.6311 | 1.4078 | 1.12× |

Table 8: Augmentation-side wall-clock latency for ICR and BA-TTA under matched local 7B–8B augmentation backbones with four stochastic rewrites per sample.

## 5.5 Sensitivity Analysis: Impact of ID Exemplars

To examine the stability of BA-TTA and verify that its gains are not overly dependent on a particular ID exemplar configuration, we conduct a sensitivity analysis on the number of in-distribution exemplars used

in the prompt. Specifically, for toxicity detection, we vary the total number of exemplars from 6 to 16 (i.e., 3 to 8 per class), and for sentiment analysis, from 9 to 21 (i.e., 3 to 7 per class). The results are reported in Table 13.

**Impact of Exemplar Quantity ($k$).** Overall, BA-TTA remains stable across a broad range of exemplar counts, and consistently outperforms the non-augmented baseline for all task models. On AC, performance improves sharply once a small number of ID exemplars is introduced, rising from 46.97%, 30.46%, and 27.79% without augmentation to 75.00%, 68.45%, and 56.19% with 6 exemplars for T5, BERT, and Llama2, respectively. Performance peaks at 10 exemplars (5 per class), reaching 76.70%, 70.75%, and 57.16%, and then fluctuates only mildly as more exemplars are added. A similar trend is observed on SST5: compared with the no-augmentation baseline of 76.12%, 68.47%, and 74.16%, BA-TTA achieves strong performance across all tested exemplar budgets, with the best results obtained at 15 exemplars (5 per class), namely 77.61%, 74.44%, and 71.55%. These results indicate that BA-TTA does not require a large or carefully tuned exemplar set; instead, a moderate number of ID examples is sufficient to provide a stable estimate of the target in-distribution background. Increasing $k$ beyond this point yields diminishing returns, and may introduce slight prompt redundancy or context inefficiency for smaller 7B–8B auxiliary models.

## 6 Limitations

While our proposed test-time domain transformation framework demonstrates promising improvements in OOD robustness, it is not without limitations. First, the reliance on LLMs for in-context augmentation introduces addition computational and financial costs, particularly when processing large-scale datasets or operating in real-time applications. Second, as observed in our toxicity experiments, domain transformation yields substantial gains on Adv. Civil, where inputs are longer, context-rich comments, but degrades performance on ToxiGen's concise, sentence-level probes. This suggests that when target samples lack broader contextual cues, the transformation may introduce noise rather than informative variation, limiting its applicability to tasks or datasets composed of very short, self-contained utterances. Additionally, the augmented representations may not always align perfectly with the intended in-distribution (ID) characteristics, especially for highly divergent domains, which could limit the generalisability of the approach. Finally, while our experiments focused on sentiment analysis and toxicity detection, further evaluation across a broader set of NLP tasks is needed to validate the robustness and scalability of the framework.

## 7 Conclusions

In this paper, we introduced BA-TTA, a test-time domain transformation framework designed to improve NLP robustness under out-of-distribution (OOD) conditions. By leveraging LLMs for background-aware, semantically constrained rewriting, our approach generates transformed test inputs whose background characteristics are brought closer to the in-distribution (ID) data while preserving label-relevant meaning. Across sentiment analysis and news topic classification benchmarks, our results show that BA-TTA consistently improves robustness over standard augmentation baselines and unconstrained in-context rewriting in many important OOD settings. For toxicity detection task, the gains are particularly strong on context-rich benchmarks such as Adv. Civil, where background-aware transformation is able to reduce substantial distribution mismatch without modifying downstream model weights.

At the same time, our results also clarify the boundary of the method. BA-TTA is most effective when the domain shift is dominated by background/style mismatch and the auxiliary LLM is sufficiently capable of carrying out semantics-preserving transformation. Under more extreme shifts, especially when inputs are short, implicit, and tightly entangled with label semantics (norm-sensitive), the limiting factor becomes whether the chosen compact augmentation model can faithfully realize the intended transformation. This distinction is supported by the strong results obtained with SB7 and GPT-4o, as well as the more mixed outcomes observed for some recent compact open-weight models on difficult toxicity benchmarks. Overall, BA-TTA provides a practical and effective black-box robustness strategy for moderately resource-constrained local settings, while also highlighting the importance of auxiliary-model capability in test-time transformation.

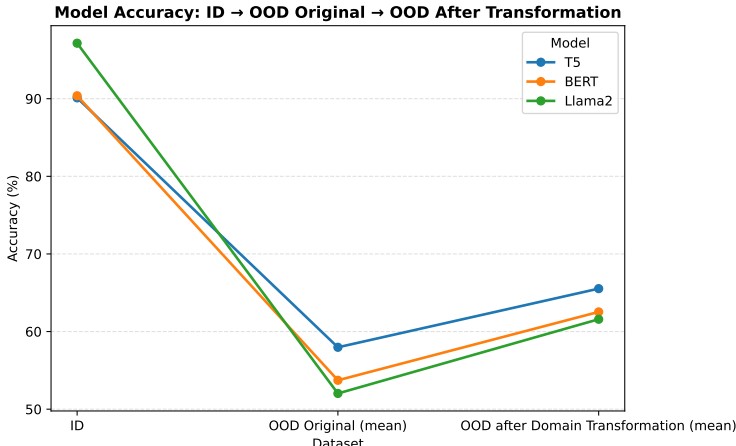

Figure 5: **OOD Performance Increase:** Comparison of T5, BERT and Llama 2 accuracies on in-distribution (ID) data, the mean out-of-distribution (OOD) splits before domain transformation, and the mean OOD splits after applying domain transformation. All three models experience a substantial performance drop when evaluated on the original OOD data, with T5 falling from 90.1% to 57.97%, BERT from 90.4% to 52.05%, and Llama 2 from 97.2% to 52.04%.

Future work may explore more sophisticated aggregation methods, stronger and more efficient auxiliary rewriters, quantized or compressed deployment for lower-resource settings, and a deeper analysis of which components of the transformation contribute most to robustness. Extending this framework to additional NLP tasks, and studying how compression, prompt design, and transformation quality interact with downstream OOD performance, would further clarify the scope and scalability of background-aware test-time augmentation.

**Acknowledgments**

This publication was supported by the Research Ireland Centre for Research Training in Artificial Intelligence under Grant No. 18/CRT/6223, the Insight Research Ireland Centre for Data Analytics under Grant No. SFI/12/RC/2289_P2, and, in part, by funding from the Canada Research Chairs Program. We gratefully acknowledge this support. We also thank our colleagues and collaborators for their valuable insights and contributions, and our families and friends for their continued encouragement throughout this work.

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

# A    Detailed Main Results

| Task Model | Augmentation Method | Sentiment Analysis | | | Toxicity Detection | | | News Topic | Summary |
|---|---|---|---|---|---|---|---|---|---|
| | | SST-5 | Dynasent | SemEval | ToxiGen | Adv. Civil | ImplicitHate | Tweets | Avg.* |
| T5 | Without augmentation | 76.12% | 47.73% | 50.07% | 65.78% | 46.97% | 63.93% | 89.01% | 62.80% |
| | Word insert | 74.35% | 46.23% | 49.49% | 66.52% | 42.11% | 64.23% | 89.96% | 61.84% |
| | Word substitute | 70.34% | 44.72% | 48.65% | 64.51% | 41.75% | 64.73% | 89.18% | 60.55% |
| | Back-Translation | 69.12% | 45.44% | 47.93% | 65.57% | 43.93% | 63.61% | 88.41% | 60.57% |
| | In-Context Rewriting | 76.12% | **52.11%** | 50.74% | 64.72% | 60.07% | **66.50%** | 90.38% | 65.81% |
| | Domain Transformation (Ours) | **77.99%** | 50.55% | **51.30%** | **66.63%** | **74.76%** | 64.37% | **91.11%** | **68.10%** |
| BERT | Without augmentation | 68.47% | 42.71% | 44.97% | 66.74% | 30.46% | 64.53% | 88.57% | 58.06% |
| | Word insert | 68.47% | 41.50% | 44.18% | **68.11%** | 27.06% | 64.92% | 88.87% | 57.59% |
| | Word substitute | 64.93% | 41.62% | 43.25% | 65.36% | 28.16% | 65.36% | 88.51% | 56.74% |
| | Back-Translation | 66.60% | 39.81% | 43.73% | 64.51% | 43.20% | 63.67% | 88.41% | 58.56% |
| | In-Context Rewriting | 72.01% | **48.63%** | 47.96% | 63.24% | 52.79% | **66.49%** | 89.51% | 62.95% |
| | Domain Transformation (Ours) | **73.69%** | 44.91% | **49.21%** | 66.42% | **67.23%** | 64.59% | **90.33%** | **65.20%** |
| Llama-2 | Without augmentation | **74.16%** | 42.75% | 39.20% | **65.89%** | 29.73% | 67.69% | 86.80% | 58.03% |
| | Word insert | 66.98% | 40.21% | 37.64% | 65.47% | 23.18% | 66.89% | 91.36% | 55.96% |
| | Word substitute | 57.93% | 38.61% | 36.20% | 62.71% | 22.69% | 65.72% | 90.43% | 53.47% |
| | Back-Translation | 65.58% | 40.49% | 39.29% | 64.20% | 36.89% | 66.31% | 90.37% | 57.59% |
| | In-Context Rewriting | 70.34% | 42.38% | 38.79% | 62.08% | 36.77% | 65.80% | 90.05% | 58.03% |
| | Domain Transformation (Ours) | 72.57% | **44.55%** | **39.49%** | 66.84% | **55.83%** | **67.94%** | **91.53%** | **62.68%** |

Table 9: **OOD performance** across various task models using the **SB7** augmentation generator.

| Task Model | Augmentation Method | Sentiment Analysis | | | Toxicity Detection | | | News Topic | Summary |
|---|---|---|---|---|---|---|---|---|---|
| | | SST-5 | Dynasent | SemEval | ToxiGen | Adv. Civil | ImplicitHate | Tweets | Avg.* |
| T5 | Without augmentation | 76.12% | 47.73% | 50.07% | **65.78%** | 46.97% | 63.93% | 89.01% | 62.80% |
| | Word insert | 74.35% | 46.23% | 49.49% | 66.52% | 42.11% | 64.23% | 89.96% | 61.84% |
| | Word substitute | 70.34% | 44.72% | 48.65% | 64.51% | 41.75% | 64.73% | 89.18% | 60.55% |
| | Back-Translation | 69.12% | 45.44% | 47.93% | 65.57% | 43.93% | 63.61% | 88.41% | 60.57% |
| | In-Context Rewriting | 75.37% | 52.31% | 50.18% | 62.18% | 48.42% | **64.19%** | 90.08% | 63.25% |
| | Domain Transformation (Ours) | **77.26%** | **53.63%** | **51.67%** | 64.72% | **78.76%** | 63.83% | **90.54%** | **68.63%** |
| BERT | Without augmentation | 68.47% | 42.71% | 44.97% | 66.74% | 30.46% | 64.53% | 88.57% | 58.06% |
| | Word insert | 68.47% | 41.50% | 44.18% | **68.11%** | 27.06% | 64.92% | 88.87% | 57.59% |
| | Word substitute | 64.93% | 41.62% | 43.25% | 65.36% | 28.16% | 65.36% | 88.51% | 56.74% |
| | Back-Translation | 66.60% | 39.81% | 43.73% | 64.51% | 43.20% | 63.67% | 88.41% | 58.56% |
| | In-Context Rewriting | 70.34% | **47.89%** | 48.04% | 62.29% | 42.96% | **64.53%** | 88.88% | 60.70% |
| | Domain Transformation (Ours) | **72.11%** | 44.98% | **48.21%** | 65.25% | **72.21%** | 64.35% | **90.64%** | **65.39%** |
| Llama-2 | Without augmentation | **74.16%** | 42.75% | 39.20% | **65.89%** | 29.73% | 67.69% | 86.80% | 58.03% |
| | Word insert | 66.98% | 40.21% | 37.64% | 65.47% | 23.18% | 66.89% | 91.36% | 55.96% |
| | Word substitute | 57.93% | 38.61% | 36.20% | 62.71% | 22.69% | 65.72% | 90.43% | 53.47% |
| | Back-Translation | 65.58% | 40.49% | 39.29% | 64.20% | 36.89% | 66.31% | 90.37% | 57.59% |
| | In-Context Rewriting | 67.19% | 42.78% | 38.97% | 59.43% | 25.73% | 64.32% | 87.29% | 55.10% |
| | Domain Transformation (Ours) | 72.01% | **45.62%** | **41.91%** | 64.62% | **57.52%** | 67.42% | **91.72%** | **63.08%** |

Table 10: **OOD performance** across various task models using **Qwen-2.5-7B-Instruct** for augmentation.

| Task Model | Augmentation Method | Sentiment Analysis | | | Toxicity Detection | | | News Topic | Summary |
|---|---|---|---|---|---|---|---|---|---|
| | | SST-5 | Dynasent | SemEval | ToxiGen | Adv. Civil | ImplicitHate | Tweets | Avg.* |
| T5 | Without augmentation | 76.12% | 47.73% | 50.07% | **65.78%** | 46.97% | 63.93% | 89.01% | 62.80% |
| | Word insert | 74.35% | 46.23% | 49.49% | 66.52% | 42.11% | 64.23% | 89.96% | 61.84% |
| | Word substitute | 70.34% | 44.72% | 48.65% | 64.51% | 41.75% | 64.73% | 89.18% | 60.55% |
| | Back-Translation | 69.12% | 45.44% | 47.93% | 65.57% | 43.93% | 63.61% | 88.41% | 60.57% |
| | In-Context Rewriting | 70.43% | 52.06% | 49.03% | 60.70% | 58.37% | 61.82% | 90.84% | 63.32% |
| | Domain Transformation (Ours) | **76.40%** | **55.14%** | **52.20%** | 62.18% | **77.31%** | 61.80% | **90.88%** | **67.99%** |
| BERT | Without augmentation | 68.47% | 42.71% | 44.97% | 66.74% | 30.46% | 64.53% | 88.57% | 58.06% |
| | Word insert | 68.47% | 41.50% | 44.18% | **68.11%** | 27.06% | 64.92% | 88.87% | 57.59% |
| | Word substitute | 64.93% | 41.62% | 43.25% | 65.36% | 28.16% | 65.36% | 88.51% | 56.74% |
| | Back-Translation | 66.60% | 39.81% | 43.73% | 64.51% | 43.20% | 63.67% | 88.41% | 58.56% |
| | In-Context Rewriting | 67.07% | 47.96% | 46.92% | 59.53% | 52.06% | 61.93% | **90.88%** | 60.91% |
| | Domain Transformation (Ours) | **73.88%** | **51.30%** | **50.63%** | 62.50% | **75.00%** | 62.11% | 90.49% | **66.56%** |
| Llama-2 | Without augmentation | **74.16%** | 42.75% | 39.20% | **65.78%** | 29.73% | **67.87%** | 86.80% | 58.04% |
| | Word insert | 66.98% | 40.21% | 37.64% | 65.47% | 23.18% | 66.89% | 91.36% | 55.96% |
| | Word substitute | 57.93% | 38.61% | 36.20% | 62.71% | 22.69% | 65.72% | 90.43% | 53.47% |
| | Back-Translation | 65.58% | 40.49% | 39.29% | 64.20% | 36.89% | 66.31% | 90.37% | 57.59% |
| | In-Context Rewriting | 71.27% | 45.74% | 42.73% | 59.85% | 31.55% | 63.73% | **91.67%** | 58.08% |
| | Domain Transformation (Ours) | 74.07% | **48.01%** | **43.53%** | 64.23% | **60.80%** | 66.22% | 91.46% | **64.05%** |

Table 11: **OOD performance** across various task models using **Llama-3.1-8B-Instruct** for augmentation.

## B    Failure Cases and Semantic Preservation Limits

To better characterize the limitations of BA-TTA, we manually inspect representative examples where the original prediction is correct but the final BA-TTA prediction becomes incorrect after rewriting. We observe three recurring failure modes. (1) *Semantic drift*: the rewrite changes label-relevant meaning by misinterpreting named entities, titles, or figurative phrasing. (2) *Hallucinated detail*: the model introduces unsupported content, such as product-review narratives, user complaints, or additional evaluative statements that do not appear in the source. (3) *Intensity or evaluation shift*: the topic remains roughly similar, but the rewrite changes the polarity, strength, or evaluative framing of the original text. Table 12 shows representative examples. These cases are especially common on short and implicit inputs, where semantics and style are tightly entangled and compact 7B–8B rewriters have limited room to adjust background features without disturbing the underlying meaning.

| Original Input | BA-TTA Rewrite | Failure Type | Observed Issue |
|---|---|---|---|
| Stay Away by Nirvana is just the perfect song to listen to tomorrow morning. | Steer clear of this album; "Stay Away" is the ideal track to play in the early morning. | Semantic drift | The song title *Stay Away* is misread as an imperative phrase, changing the meaning of the original statement. |
| Army cuts off Boko Haram's food supply: The Nigerian Army said Thursday it has succeeded in cutting off the supply chain . . . | The product description clearly states that the package includes everything needed to start using the item right away . . . | Hallucinated detail | The rewrite collapses into an unrelated product-review domain and invents content absent from the source. |
| Got the feeling that if Ocarina of Time is the 1st Star Wars trilogy, then Wind Waker is the prequels and this new one is Twilight Princess. | If Ocarina of Time feels like the original Star Wars trilogy, then Wind Waker resembles the prequels . . . however, this latest game might not quite reach the heights of the previous titles. | Evaluation shift | The rewrite injects a negative evaluative judgment not present in the original text. |
| Murray going for more swear words than South Park the movie! C'mon Andy you can turn this around! #usopen #passion | Murray's language is excessive, bordering on offensive! Come on Andy, you can improve your game! #usopen #enthusiasm | Intensity shift | The rewrite changes the emotional and evaluative framing, which can alter downstream sentiment prediction. |
| @robbieJBhood Kane walked off with a tight calf last Saturday. If he starts I guess N'jie will be on the bench. | The product arrived with a slight defect last Saturday. If it functions properly, the replacement part will need to be on the bench. | Hallucinated detail | A sports update is rewritten into a product-review scenario, introducing unsupported and irrelevant content. |
| I don't think Nicki's pregnant. Stomach was flat again in the side view. Think she was just sticking her butt out in the 1st pic. | I don't believe the package is filled. In the side view, the contents appeared flat again. It seems like the item was just shifted in the first image. | Semantic drift | The rewrite preserves superficial structure but replaces the original meaning with an unrelated package/product interpretation. |

Table 12: Representative failure cases of BA-TTA. Despite semantic constraints, compact 7B–8B rewriters may still introduce semantic drift, hallucinated detail, or evaluative/intensity shifts, especially under difficult domain shifts.

## C   Sensitivity Analysis Table: Impact of ID Exemplars Number

| Dataset | Configuration (SB7) | T5 | BERT | Llama |
|---|---|---|---|---|
| **AC - Toxicity** | Without Augmentation | 46.97% | 30.46% | 27.79% |
| | 6 ID exemplars (3 each class) | 75.00% | 68.45% | 56.19% |
| | 8 ID exemplars (4 each class) | 75.61% | 68.93% | 55.83% |
| | **10 ID exemplars (5 each class)** | **76.70%** | **70.75%** | **57.16%** |
| | 12 ID exemplars (6 each class) | 75.24% | 70.27% | 55.70% |
| | 14 ID exemplars (7 each class) | 75.12% | 68.93% | 56.31% |
| | 16 ID exemplars (8 each class) | 74.76% | 66.49% | 55.83% |
| **SST5 - Sentiment** | Without Augmentation | 76.12% | 68.47% | 74.16% |
| | 9 ID exemplars (3 each class) | 75.93% | 73.60% | 71.46% |
| | 12 ID exemplars (4 each class) | 76.87% | 72.95% | 71.08% |
| | **15 ID exemplars (5 each class)** | **77.61%** | **74.44%** | **71.55%** |
| | 18 ID exemplars (6 each class) | 77.24% | 73.88% | 71.46% |
| | 21 ID exemplars (7 each class) | 77.52% | 73.32% | 71.36% |

Table 13: Sensitivity analysis of ID exemplar counts across Toxicity and Sentiment datasets. Performance (%) is reported for T5, BERT, and Llama-2 models. Bold values indicate the optimal exemplar count.

## D   Sample Data

| No. | Sample |
|---|---|
| 1 | we should build a wall on the border to keep them from coming in illegally. |
| 2 | having a disability means you can get everything for free |
| 3 | they are taking all of our jobs |
| 4 | the return of people of color to the spaces where they were once enslaved |
| 5 | iran is not the same thing as isis |

Table 14: Samples from ToxiGen dataset

| No. | Sample |
|-----|--------|
| 1 | What a stuppid thing to say. TTake a course on trading. By your logic markets would stay at the same price in perpetuity. I can assure you that at this poinyt in the cycle it will take surprisingly little to go from a buyers market to a sellers market. When the sellers overpower buyers you'll see prices tank. Te fact that I have to even explani this elementary concept reminds me thaft there is no fix for stupi.d The former alignment of Route 29 in the northern part |
| 2 | I misspoke the way I should have said it" Political speech for, "I'm an idilt and I got caught in my lie to one part of the country by another part of the country." I was just happy to see Calgary booing our selfie seeking PM. Some where in Quebec there is a village that ks missing it's idiiot and we need to get him back tto them before he completely ruins this country. |
| 3 | Anoter cause of the sewage spills rather than "human error" could be better attributed to stuppidity, Incompetence, negligence. I suppose it is kind of late in the game since Capt Kirk well into his second and llast term as Mayor but maybe not too late to fire the environmental services director who seems to be a prime example oof the Peter principle, you know rising to the level of your incompetence. A further example of Starr 's allegedly anti |

Table 15: Samples from Adv.Civil dataset

